# Mentoring and Monitoring of Student Teachers in Their In-School Placements—The Case of the University of Santiago de Compostela

**Carmen Fernández-Morante** [1,*] **, Martin Manuel Leránoz-Iglesias** [2] **, Beatriz Cebreiro-López** [1] **and Cristina Abeal-Pereira** [1]

1   Department of Pedagogy and Didactics, University of Santiago de Compostela,
    15705 Santiago de Compostela, Spain; beatriz.cebreiro@usc.es (B.C.-L.); cristina.abeal@usc.es (C.A.-P.)
2   Department of Applied Didactics, University of Santiago de Compostela,
    15705 Santiago de Compostela, Spain; martin.leranoz@usc.es
*   Correspondence: carmen.morante@usc.es

**Abstract:** (1) The in-school placement is a key scenario in linking theory and practice in the processes of initial teacher education (ITE) and constitutes a crucial element in the understanding of professional competencies and the experimentation of educational innovation proposals. For this reason, the pedagogical model that guides teaching practices and the set of learning activities that university and school mentors develop is especially relevant. This paper aims to analyze the current individualized monitoring processes of student teachers, as well as to detect difficulties and needs for improvement. (2) The center of interest corresponds to the internship mentors of the ITE degrees of the Faculty of Education Sciences at the University of Santiago de Compostela. This research was carried out through a descriptive study with a survey method and a study sample of 202 mentors. (3) Results show that there are discrepancies in the activities that university and school mentors implement with students, as well as in the observation and supervision processes. (4) We propose a learning reflective methodology based on the Technological Pedagogical Content Knowledge (TPACK) model to coordinate agents involved in the internship, connecting them through a specialized eLearning environment equipped with specific tools to facilitate the processes of individualized monitoring, reflective learning, and self-regulation.

**Keywords:** initial teacher education; in-school placements; reflective learning; student teachers; TPACK; eLearning

## 1. Introduction

The research is an integral component of the accredited university programs that prepare individuals for the teaching profession at the University of Santiago de Compostela. Its objective is to conduct an examination of the procedures involved in individually monitoring student teachers during their internships, specifically from the perspective of their mentors. The study is part of the International Competitive Research Project EKT—Educational Knowledge Transfer (reference no. 612414-EPP-1-2019-1-EN-EPPKA2-KA). This European Knowledge Alliance aimed to develop and experiment with collaborative educational methodologies and an intelligent technological system to improve initial teacher education (ITE) by building bridges between professional and academic contexts.

In-school placements (ISPs) cover approximately 25% of the training modules of official university degrees in education that enable the exercise of the teaching profession in Spain, which is a key scenario for understanding and exercising professional skills and experimenting with educational innovation proposals supervised by experienced professionals with the support of educational researchers (Fernández-Morante et al. 2021). These internships immerse students in the professional educational field with the aim of

"learning to teach", articulating theoretical knowledge with practical knowledge, analyzing, and understanding their relationships for the acquisition of a critical and reflective ITE (Brandão and Fernández-Morante 2023; Zabalza 2014). The training of students in the curricular subjects that make up the degrees taught in the Faculties of Education and the learning that is mobilized in the internships in the placement schools are usually a simile of what is understood by theory and practice. Theory is the knowledge needed to face professional activity through the different curricular subjects of ITE, and professional practice is understood as a range of opportunities for observation, analysis, interpretation of what happens, how and when it happens in teaching situations, and experimentation in real contexts. In this way, the university provides knowledge through its teaching staff in the teaching professions that are mobilized in the Faculties of Education, while the placement schools contextualize, apply, and recreate it in a practical way in their classrooms (Montero Mesa 2018).

A fundamental aspect of the in-school placement is to focus on the relationship between the institutions involved in it since they have different cultures, and their shared responsibility in this process has been limited to a strategy of minimal contact and "temporary cession" of developmental responsibility during the short time that students are in schools. Taking this situation into account, it is necessary to establish effective bridges of communication and collaboration between the agents that accompany student teachers in their training. This accompaniment and monitoring of students should be a shared, collaborative task and joint work between their mentors, promoting co-participatory reflection for the development of skills and acquisition of competencies of student teachers (Leránoz-Iglesias 2023). A noteworthy factor is the enrichment of the intervention of numerous agents and media throughout this educational process, favoring the construction of a personal project and a way of understanding education (Puig 2006). The mentors in universities and placement schools, together with the students, form a "triad" that houses a series of previous beliefs and assumptions that feed back into each other, acting as a filter and guide for the actions toward the other members included in it (González and Fuentes 2011). The mentoring process for student teachers requires the definition of the roles of both mentors, their functions, and tasks, as well as the planning of collaborative and shared work throughout the process. Some studies define the role of the university mentor (also known as an academic mentor or university supervisor) as a distant accompaniment outside of the educational action, whose main functions are based on the guidance, planning, motivation, and reflective research of the students (Castaño et al. 1997; Puig 2004). The school mentors (also called professional or technical mentors) define their role as daily, close, and direct to the students, and their reference in placement school is to generate immediate learning through observation and direct experience in the context itself. The school mentor must possess knowledge, leadership, and personal skills to develop the functions of reception, accompaniment, and guidance in the dynamic participation of the placement school/classroom and the assessment of the student, where co-participatory planning and action of the mentors of both contexts is deemed necessary (Cid et al. 2011; Díaz and Díaz 2012). However, different studies (Leránoz-Iglesias 2023; López 2017; Megía 2016; Orland-Barak and Wang 2021; Porto and Martínez 2013) show that there is little coordination and collaboration between the mentors of both institutions, which may be due, among others, to the difference in teaching profiles, the training on mentoring, the degree of involvement and motivation, the time spent, and little or no recognition. These aspects have a negative influence on the supervision and tasks they have to carry out with the student teachers in school practice. The lack of coordination and joint work between them and the lack of communication in the clarification of the functions and activities to be carried out by each of them leads to heterogeneous mentoring, generally assuming traditional, individualistic, and uncooperative models. Therefore, the in-school placement requires a training system that uses multiple educational resources and technologies aligned with its specificity, implementing innovative experiences and expanding its functions, elaborating a new role in mentoring, with work dynamics that improve communication between and

within participants, individualized monitoring, reflective thinking, and formative and continuous student assessment (Cebrián-de-la-Serna and Cebrián-Robles 2023; Imbernón 2016; Leránoz-Iglesias et al. 2023; Saiz-Linares and Ceballos-López 2019).

Over the last two decades, ICT has been progressively introduced as an alternative to traditional resources in teacher education. However, research into educational technology has noted problems in its development, such as excessive technification and the absence of conceptual models available to the educational community for teacher education. For this reason, it is necessary to establish a balance where educational technology is a key element in the ISP methodology, integrating its pedagogical content and its own disciplinary content, in short, following the model under which some authors (Balladares-Burgos and Valverde-Berrocoso 2022; Cabero and Barroso 2016; Mishra and Koehler 2006) call this hybridization: TPACK, technological pedagogical knowledge of content. Technological knowledge involves the competence that teachers possess in the implementation and combination of technological tools, both general and specific. Pedagogical knowledge refers to the processes, strategies, and methods of teaching activities, with knowledge of the content being the teacher's mastery of the discipline. In this sense, the hybridization provided by TPACK can provide the necessary balance to achieve true integration of ICT, addressing the methodology and developing dynamic and adequate solutions to the needs of the student teachers.

The purpose of this study is to carry out an analysis of the processes of individualized monitoring of students in internships, as well as to detect difficulties and needs for improvement from the point of view of university and school mentors. This knowledge will allow technological and pedagogical decisions to be made aimed at the development of useful digital tools to mobilize and enhance the learning expected from the internship period. To this end, we analyze, on the one hand, the frequency of activities that mentors work with the students during the phases of the internship, before, during, and after the stay in the placement school and, on the other hand, the frequency of observation and supervision on the aspects that the students must develop during their internships. The study is complemented by contributions from mentors through suggestions and proposals to improve the process of individualized follow-up of student teachers' trainees, as well as the contribution of resources and technological tools that can be useful to promote and benefit this process. The results allowed proposals to be formulated for the design of the EKT eLearning system.

## 2. Materials and Methods

The research opted for an interpretative approach using the type of descriptive study framed in cross-sectional observational research models. The design carried out corresponds to a non-experimental study of quantitative methodology using the questionnaire as an instrument to obtain data.

### 2.1. Participants

An intentional sampling was carried out among the total number of teachers who had assumed the role of university and school mentors during the 2019/2020 and 2020/2021 academic years in any of the in-school placement subjects of the Bachelor's Degree in Early Childhood Education, Primary Education and the Master's Degree in Teaching of Compulsory Secondary Education and Baccalaureate, Vocational Training, and Language Teaching from the Faculty of Education Sciences of the University of Santiago de Compostela. The sample consisted of a total of N = 202 informants, assuming a confidence level of 95% and a sampling error of 6%. A total of 65.8% (N = 133) of the informants were women, as is usual considering the feminization in teaching studies and the teaching profession. The average age of the study participants was 49 years and ranged from 27 to 68 years, with an average teaching experience of 21, 59 years between 2 and 42 years. The average number of years of experience as an internship mentor for both respondent profiles (university and school) was 7.89 years and ranged from 0 to 34 years of experience. By profile, 22.3% were

from a university and 77.7% from a school, a distribution consistent with the distribution of both profiles in the in-school placement since university mentors usually supervise multiple student teachers' trainees simultaneously. Disaggregating the profile of school mentors, 32.2% corresponded to teachers of initial education (Early Childhood and Primary Education) and 45.5% to secondary school teachers (Compulsory Secondary Education, Baccalaureate, and Vocational Training).

### *2.2. Instrument*

In the first phase of the project implementation, the EKT questionnaire was developed for internship mentors of both profiles (university and school). The questionnaire is made up of a total of 62 items organized into 5 sections and three study dimensions. The instrument was validated prior to its application and had a high reliability index calculated from Cronbach's alpha ($\alpha$ = 0.970). As previously mentioned, this paper presents the findings obtained in the second dimension of the study, whose purpose was to analyze and detect difficulties and needs for improvement of the processes of individualized follow-up of students in internships from the point of view of their mentors. The EKT questionnaire for this dimension incorporated nominal multiple-choice questions to find out the type and frequency of activities and/or processes that mentors carried out with student teachers' trainees and the frequency of observation and supervision of key professional skills and activities during the internship training of student teachers. Open questions were also incorporated to provide proposals and suggestions for improving the individualized monitoring of students in internships, as well as tools and resources that they considered useful for the improvement of the training process.

### *2.3. Procedure and Data Analysis*

To obtain data, the questionnaire was distributed in the two official languages of the Autonomous Community of Galicia (Galician and Spanish) and online. Data were collected over 3 months using the institutional software of the USC (Microsoft Forms version 16.66.1) under the protection of the ethical standards of research and data protection of the University of Santiago de Compostela. The study was supported by the USC Research Bioethics Committee. Data analysis was performed using IBM's *SPSS Statistics* software (version 27) under the software license of the University of Santiago de Compostela. For data analysis, descriptive frequency statistics and contrast statistics were used, applying Student's *t*-test for independent samples, taking the significance level below 0.05. Dependent variables were taken as those that refer to the teaching profile of the mentor, either from universities or placement schools and, in this latter case, as school mentors of initial or secondary education. Coding and content analysis procedures were also performed on the data from the open questions. The research design and procedures were authorized by the USC Research Ethical Committee (November 2020. Code: USC-23/2020).

### 3. Results

Using international experts (from 5 ITE European institutions members of the EKT consortium), the themes and key elements around which the process of observation and action of student teachers during the internships should revolve were defined, and the frequency with which they were approached by the mentors in their accompaniment process was analyzed. Figure 1 shows the frequency with which these key elements were addressed by the mentors in the different phases of the process (before arriving at the placement schools, during their stay there, and in the closing stage of the process that takes place on their return to university).

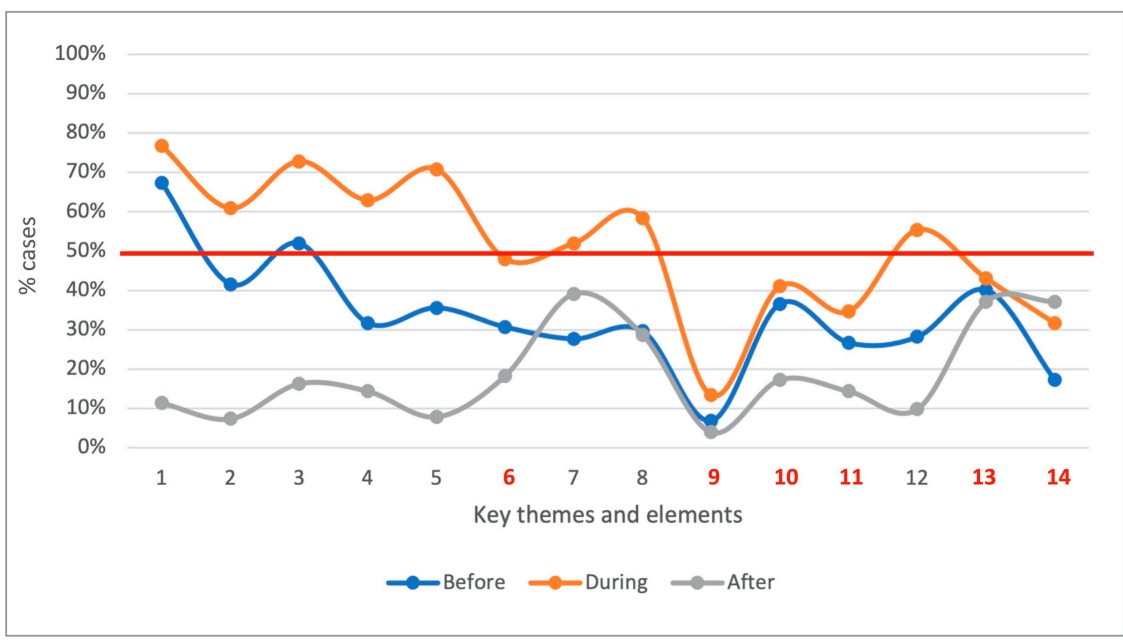

**Figure 1.** Key themes and elements that mentors work on with student teachers during internship.

It is observed that, in general, there is a very uneven distribution of the intensity with which the different themes are addressed in the three phases of the training. Thus, there is a concentration during the placement school stay (in which the school mentor is mainly involved) and a lower frequency in the preparation and closing phases in which university mentors mainly intervene. It is also observed that more than half of the mentors do not address at any point in the supervision process some of the key themes and elements to which the attention of student teachers should be directed during the internships for their reflective learning. This is the case for aspects highlighted in red (Figure 1) relating to the following:

1. Theme 6: Design, development, or adaptation of assessment instruments (before: 18.35%, during: 30.79%, after: 48%);
2. Theme 9: Feedback to families (before: 4%, during: 6.9%, after: 13.4%);
3. Theme 10: Communication with other teaching, coordination, or management staff (before: 17.3%, during: 36.6%, after: 41.1%);
4. Theme 11: Meeting with other subject/department teaching staff (before: 14.40%, during: 26.70%, after: 34.70%);
5. Theme 13: Coordination of the student's training plan (before: 37.10%, during: 37.10%, after: 43.10%);
6. Theme 14: Preparation of progress reports for pupils (before: 17.30%, during: 31.70%, after: 37.10%).

Table 1 shows, by profile, the extent to which the internship mentors address the key themes and elements in each of the phases of the in-school placement, as well as where the attention of student teachers should be during reflective learning. If we analyze the data relating to the approach to the issues in the first stage of the process, i.e., in the preparation phase prior to arrival at the placement schools, it can be said that they work with little intensity in this phase. Only two of them are addressed by more than half of the mentors: classroom teaching planning (67.30% of the cases) and the selection of materials and resources for teaching practice (52% of the cases). It could be said that in this first phase, the process of supervision and advice of the mentors is limited to issues related to the organization of the process (assignment of placement center and mentor, rules, and procedures, preparation of the stay, etc.) and that the key aspects of the educational experience that they will observe in real contexts are hardly addressed in this phase.

**Table 1.** Descriptive statistics on the themes and key elements that university and school mentors work on with student teachers in each phase of the internship.

| Item | Activities | Mentor | N | Before<br>% Cases | During<br>% Cases | After<br>% Cases |
|------|-----------|--------|---|---------|---------|---------|
| 1 | Planning classroom teaching | University | 45 | 57.80 | 64.40 | 8.90 |
| | | Schools | 157 | 70.10 | 80.30 | 12.10 |
| | | Total | 202 | 67.30 | 76.70 | 11.40 |
| 2 | Classroom organization | University | 45 | 20.00 | 33.30 | 4.40 |
| | | Schools | 157 | 47.80 | 68.80 | 8.30 |
| | | Total | 202 | 41.60 | 60.90 | 7.40 |
| 3 | Selection of materials/resources for teaching practice | University | 45 | 42.20 | 57.80 | 11.10 |
| | | Schools | 157 | 54.80 | 77.10 | 17.80 |
| | | Total | 202 | 52.00 | 72.80 | 16.30 |
| 4 | Production of learning materials and resources | University | 45 | 15.60 | 44.40 | 4.40 |
| | | Schools | 157 | 36.30 | 68.20 | 17.20 |
| | | Total | 202 | 31.70 | 62.90 | 14.40 |
| 5 | Direct teaching | University | 45 | 13.30 | 13.30 | 2.20 |
| | | Schools | 157 | 42.30 | 87.30 | 9.60 |
| | | Total | 202 | 35.60 | 70.80 | 7.90 |
| 6 | Design, development, or adaptation of assessment instruments | University | 45 | 28.90 | 42.20 | 17.80 |
| | | Schools | 157 | 31.20 | 49.70 | 18.50 |
| | | Total | 202 | 30.70 | 48.00 | 18.30 |
| 7 | Student assessment | University | 45 | 20.00 | 24.40 | 51.10 |
| | | Schools | 157 | 29.90 | 59.90 | 35.70 |
| | | Total | 202 | 27.70 | 52.00 | 39.10 |
| 8 | Feedback to students | University | 45 | 24.40 | 44.40 | 44.40 |
| | | Schools | 157 | 31.20 | 62.40 | 24.20 |
| | | Total | 202 | 29.70 | 58.40 | 28.70 |
| 9 | Feedback to families | University | 45 | 4.40 | 4.40 | 2.20 |
| | | Schools | 157 | 7.60 | 15.90 | 4.50 |
| | | Total | 202 | 6.90 | 13.40 | 4.00 |
| 10 | Communication with other teaching, coordination, or management staff | University | 45 | 24.40 | 20.00 | 26.70 |
| | | Schools | 157 | 40.10 | 47.10 | 14.60 |
| | | Total | 202 | 36.60 | 41.10 | 17.30 |
| 11 | Meeting with other subject/department faculty | University | 45 | 20.00 | 20.00 | 26.70 |
| | | Schools | 157 | 28.70 | 38.90 | 10.80 |
| | | Total | 202 | 26.70 | 34.70 | 14.40 |
| 12 | Support for students with special educational needs | University | 45 | 17.80 | 31.10 | 6.70 |
| | | Schools | 157 | 31.20 | 62.40 | 10.80 |
| | | Total | 202 | 28.20 | 55.40 | 9.90 |
| 13 | Coordination of the internship plan | University | 45 | 42.20 | 48.90 | 28.90 |
| | | Schools | 157 | 39.50 | 41.40 | 39.50 |
| | | Total | 202 | 40.10 | 43.10 | 37.10 |
| 14 | Preparation of student teacher progress reports | University | 45 | 24.40 | 35.38 | 20.47 |
| | | Schools | 157 | 15.30 | 56.20 | 18.34 |
| | | Total | 202 | 17.30 | 51.56 | 18.81 |

If we now focus on the second stage of the process, i.e., the stay of student teachers in schools, we can see that the elements related to the educational experience and the teaching function acquire greater weight. This is the case for elements such as classroom teaching planning (76.70%), selection of materials and resources (72.80%), direct classroom teaching (70.80%), production of learning materials and resources (62.90%), classroom organization (60.9%) or feedback to students (58.40%). It could, therefore, be said that it is fundamentally

in the interaction with the school mentor that these elements are addressed. There are also shortcomings in the approach to some key elements, such as those related to the processes and instruments of pupil assessment, interaction with families, and coordination with other teachers, as more than half of the mentors do not work with student teachers.

In relation to the final phase of the process, that is, once the student teachers return to their universities, the results obtained show that, in general, in this phase of the accompaniment process, the key themes and elements on which the attention of student teachers should be directed during the reflective learning are not taken up. None of the formulated elements reaches a minimum score of 50% of the cases. It could be said that in this final phase, the process of supervising and advising mentors is limited to issues relating to closure and formal evaluation of the process.

The study revealed the existence of significant differences, the intensity with which university and school mentors address the key themes and elements in the accompaniment process, and the prominence of each profile in each of the three phases. We start from the idea that the mentoring process in all its phases should be collaborative and horizontal and that all the key elements should be addressed in the three phases by the mentors with greater or lesser intensity. Thus, for example, in the preparatory phase (before), in which university mentors have a greater role, they should be formulated, thus helping the student teacher to focus his or her attention on them. In the placement school stay (during), where the school mentor plays a greater role, they should be observed, accompanying the student teachers and providing them with a broad vision of school life. Finally, in the closing phase (after), in which the student returns to the ITE institution (Faculty of Education) and in which the university mentor regains prominence, key elements should be taken up again to consolidate learning.

Table 2 shows that there are significant differences in the intensity with which the two profiles address the key elements during the process. Thus:

1. In the preparation phase, school mentors work more intensively on three of the elements: those related to the organization of the classroom (item 2 sig = 0.001 < 0.05), the production of learning materials and resources (item 2 sig = 0.008 < 0.05) and direct teaching in the classroom (item 2 sig = 0.000 < 0.05);
2. The school mentors work more intensively in the internship phase practically all the key elements: item 1 (sig = 0.027 < 0.05), item 2 (sig = 0.001 < 0.05), item 3 (sig = 0.010 < 0.05), item 4 (sig = 0.004 < 0.05), item 5 (sig = 0.001 < 0.05), item 7 (sig = 0.001 < 0.05), item 8 (sig = 0.031 < 0.05), item 9 (sig = 0.046 < 0.05), item 10 (sig = 0.001 < 0.05), item 11 (sig = 0.019 < 0.05), item 12 (sig = 0.001 < 0.05), and item 14 (sig = 0.014 < 0.05);
3. University mentors work more intensively in the closing phase on three of the key elements: those related to student feedback (item 8 sig = 0.008 < 0.05), meetings between subject and departmental teachers (item 11 sig = 0.007 < 0.05), and the preparation of student progress reports (item 14 sig = 0.028 < 0.05).

**Table 2.** Student's *t*-test on the key themes and elements that university and school mentors work on with student teachers in each phase of the internship.

| Item | Before | Mentor | N | Mean | Statis. | gl | *p* |
|------|--------|--------|---|------|---------|----|----|
| 2 | Classroom organization | University<br>School | 45<br>157 | 0.2000<br>0.4777 | −3.411 | 200 | 0.001 |
| 4 | Production of learning materials and resources | University<br>School | 45<br>157 | 0.1556<br>0.3631 | −2.671 | 200 | 0.008 |
| 5 | Direct teaching | University<br>School | 45<br>157 | 0.1333<br>0.4204 | −3.642 | 200 | 0.000 |

**Table 2.** *Cont.*

| Item | During | Mentor | N | Mean | Statis. | gl | *p* |
|---|---|---|---|---|---|---|---|
| 1 | Planning classroom teaching | University | 45 | 0.6444 | −2.229 | 200 | 0.027 |
| | | School | 157 | 0.8025 | | | |
| 2 | Classroom organization | University | 45 | 0.3333 | −4.486 | 200 | 0.001 |
| | | School | 157 | 0.6879 | | | |
| 3 | Selection of materials/resources for teaching practice | University | 45 | 0.5778 | −2.593 | 200 | 0.010 |
| | | School | 157 | 0.7707 | | | |
| 4 | Production of learning materials and resources | University | 45 | 0.4444 | −2.950 | 200 | 0.004 |
| | | School | 157 | 0.6815 | | | |
| 5 | Direct teaching | University | 45 | 0.1333 | −12.991 | 200 | 0.001 |
| | | School | 157 | 0.8726 | | | |
| 7 | Student assessment | University | 45 | 0.2444 | −4.367 | 200 | 0.001 |
| | | School | 157 | 0.5987 | | | |
| 8 | Feedback to students | University | 45 | 0.4444 | −2.171 | 200 | 0.031 |
| | | School | 157 | 0.6242 | | | |
| 9 | Feedback to families | University | 45 | 0.0444 | −2.005 | 200 | 0.046 |
| | | School | 157 | 0.1592 | | | |
| 10 | Communication with other teaching, coordination, or management staff | University | 45 | 0.2000 | −3.334 | 200 | 0.001 |
| | | School | 157 | 0.4713 | | | |
| 11 | Meeting with other subject/department faculty | University | 45 | 0.2000 | −2.364 | 200 | 0.019 |
| | | School | 157 | 0.3885 | | | |
| 12 | Support for students with special educational needs | University | 45 | 0.3111 | −3.841 | 200 | 0.001 |
| | | School | 157 | 0.6242 | | | |
| 14 | Preparation of student progress reports | University | 45 | 0.4667 | 2.475 | 200 | 0.014 |
| | | School | 157 | 0.2739 | | | |
| **Item** | **After** | **Mentor** | **N** | **Mean** | **Statis.** | **gl** | ***p*** |
| 4 | Production of learning materials and resources | University | 45 | 0.0444 | −2.165 | 200 | 0.032 |
| | | School | 157 | 0.1720 | | | |
| 8 | Feedback to students | University | 45 | 0.4444 | 2.680 | 200 | 0.008 |
| | | School | 157 | 0.2420 | | | |
| 11 | Meeting with other subject/department faculty | University | 45 | 0.2667 | 2.706 | 200 | 0.007 |
| | | School | 157 | 0.1083 | | | |
| 14 | Preparation of student progress reports | University | 45 | 0.5111 | 2.218 | 200 | 0.028 |
| | | Center | 157 | 0.3312 | | | |

Table 3 presents the data obtained by disaggregating school mentors into two categories. On the one hand, those who work in initial education (N = 65), teachers in preschool and primary education, and on the other, those who work in secondary education (N = 92), secondary school teachers, Baccalaureate and Vocational Training). As you can see, the data follow a very similar logic to those described above, so an unequal distribution is found in terms of the intensity with which the different elements are addressed in the three phases of the in-school placement, concentrating their approach during the placement phase. The data disaggregated by type of teaching also show that it is the secondary school mentors who deal with a greater variety and breadth of elements while accompanying student teachers. Thus, more than 60% of teachers in Compulsory Secondary Education, Baccalaureate, and Vocational Training direct the attention of student teachers to the following elements:

1.  Classroom teaching planning (84.78%);
2.  Classroom organization (70.65%);
3.  Selection of materials and resources (82.61%);
4.  Production of learning materials and resources (64.13%);

5.  Direct teaching in the classroom (89.13%);
6.  Design and adaptation of assessment instruments (60.87%);
7.  Student assessment (75%);
8.  Student feedback (60.87%);
9.  Communication and coordination with other teachers (53.26%);
10. Support for students with special educational needs (56.52%).

**Table 3.** Descriptive statistics on the key themes and elements that school mentors work on with student teachers in each phase of the internship according to educational level.

| Item | Elements | Teaching Mentor | N | Before % Cases | During % Cases | After % Cases |
|---|---|---|---|---|---|---|
| 1 | Planning classroom teaching | Initial | 65 | 58.46 | 73.85 | 10.77 |
|   |                             | Secondary | 92 | 78.26 | 84.78 | 13.04 |
| 2 | Classroom organization | Initial | 65 | 41.54 | 66.15 | 06.15 |
|   |                        | Secondary | 92 | 52.17 | 70.65 | 09.78 |
| 3 | Selection of materials/resources for teaching practice | Initial | 65 | 36.92 | 69.23 | 10.77 |
|   |                                                        | Secondary | 92 | 67.39 | 82.61 | 22.83 |
| 4 | Production of learning materials and resources | Initial | 65 | 26.15 | 73.85 | 07.69 |
|   |                                                | Secondary | 92 | 43.48 | 64.13 | 23.91 |
| 5 | Direct teaching | Initial | 65 | 27.69 | 84.62 | 6.15 |
|   |                 | Secondary | 92 | 52.17 | 89.13 | 11.96 |
| 6 | Design, development, or adaptation of assessment instruments | Initial | 65 | 15.38 | 33.85 | 10.77 |
|   |                                                              | Secondary | 92 | 42.39 | 60.87 | 23.91 |
| 7 | Student assessment | Initial | 65 | 15.38 | 38.46 | 23.08 |
|   |                    | Secondary | 92 | 40.22 | 75.00 | 44.57 |
| 8 | Feedback to students | Initial | 65 | 24.62 | 64.62 | 15.38 |
|   |                      | Secondary | 92 | 35.87 | 60.87 | 30.43 |
| 9 | Feedback to families | Initial | 65 | 7.69 | 16.92 | 3.08 |
|   |                      | Secondary | 92 | 7.61 | 15.22 | 5.43 |
| 10 | Communication with other teaching, coordination, or management staff | Initial | 65 | 26.15 | 38.46 | 9.23 |
|    |                                                                      | Secondary | 92 | 50.00 | 53.26 | 18.48 |
| 11 | Meeting with other subject/department faculty | Initial | 65 | 15.38 | 21.54 | 6.15 |
|    |                                               | Secondary | 92 | 38.04 | 51.09 | 14.13 |
| 12 | Support for students with special educational needs | Initial | 65 | 27.69 | 70.77 | 10.77 |
|    |                                                     | Secondary | 92 | 33.70 | 56.52 | 10.87 |
| 13 | Coordination of own training plan | Initial | 65 | 38.46 | 41.54 | 43.08 |
|    |                                   | Secondary | 92 | 40.22 | 41.30 | 36.96 |
| 14 | Preparation of student progress reports | Initial | 65 | 9.23 | 16.92 | 27.69 |
|    |                                         | Secondary | 92 | 19.57 | 34.78 | 36.96 |

If we now focus on early childhood mentors, they mostly direct the attention of student teachers to the following elements:

1.  Classroom teaching planning (73.85%);
2.  Classroom organization (66.15%);
3.  Selection of materials and resources (69.23%);
4.  Production of learning materials and resources (73.85%);
5.  Direct teaching in the classroom (84.62%);
6.  Student feedback (64.62%).

The results obtained also highlight the existing shortcomings in the approach of some key elements while accompanying student teachers. They include the design and adaptation of learning materials and resources or assessment instruments, the evaluation

of students, or feedback to families in the educational process. More than half of the school mentors report not addressing them during the internship.

The study revealed the existence of significant differences in the intensity with which school mentors address the key themes and elements in the accompaniment process according to educational levels (initial or secondary). Table 4 confirms that the mentors of Compulsory Secondary Education, Baccalaureate, and Vocational Training are the ones who address the key issues with greater intensity in the three phases of the process. This is as follows:

1.   In the preparation phase, item 1 (sig = 0.007 < 0.05), item 3 (sig = 0.000 < 0.05), item 4 (sig = 0.026 < 0.05), item 5 (sig = 0.002 < 0.05), item 6 (sig = 0.000 < 0.05), item 7 (sig = 0.001 < 0.05), item 10 (sig = 0.003 < 0.05), and item 11 (sig = 0.002 < 0.05);
2.   During the placement: item 3 (sig = 0.050 < 0.05), item 6 (sig = 0.001 < 0.05), item 7 (sig = 0.000 < 0.05), item 11 (sig = 0.000 < 0.05), and item 14 (sig = 0.013 < 0.05).
3.   In the closing phase: item 4 (sig = 0.008 < 0.05), item 6 (sig = 0.037 < 0.05), item 7 (sig = 0.005 < 0.05), and item 8 (sig = 0.030 < 0.05).

**Table 4.** Student's *t*-test on the key themes and elements that school mentors work on with student teachers in each phase of the internship depending on educational level.

| Item | Before | Mentor | N | Mean | Statis. | gl | *p* |
|---|---|---|---|---|---|---|---|
| 1 | Planning classroom teaching | Initial | 65 | 0.5846 | −2.713 | 155 | 0.007 |
| | | Secondary | 92 | 0.7826 | | | |
| 3 | Selection of materials/resources for teaching practice | Initial | 65 | 0.3692 | −3.937 | 155 | 0.000 |
| | | Secondary | 92 | 0.6739 | | | |
| 4 | Production of learning materials and resources | Initial | 65 | 0.2615 | −2.245 | 155 | 0.026 |
| | | Secondary | 92 | 0.4348 | | | |
| 5 | Direct teaching | Initial | 65 | 0.2769 | −3.136 | 155 | 0.002 |
| | | Secondary | 92 | 0.5217 | | | |
| 6 | Design, development, or adaptation of assessment instruments | Initial | 65 | 0.1538 | −3.731 | 155 | 0.000 |
| | | Secondary | 92 | 0.4239 | | | |
| 7 | Student assessment | Initial | 65 | 0.1538 | −3.450 | 155 | 0.001 |
| | | Secondary | 92 | 0.4022 | | | |
| 10 | Communication with other teaching staff, coordination, or management | Initial | 65 | 0.2615 | −3.073 | 155 | 0.003 |
| | | Secondary | 92 | 0.5000 | | | |
| 11 | Meeting with other subject/department faculty | Initial | 65 | 0.1538 | −3.171 | 155 | 0.002 |
| | | Secondary | 92 | 0.3804 | | | |
| **Item** | **During** | **Mentor** | **N** | **Mean** | **Statis.** | **gl** | ***p*** |
| 3 | Selection of materials/resources for teaching practice | Initial | 65 | 0.6923 | −1.976 | 155 | 0.050 |
| | | Secondary | 92 | 0.8261 | | | |
| 6 | Design, development, or adaptation of assessment instruments | Initial | 65 | 0.3385 | −3.438 | 155 | 0.001 |
| | | Secondary | 92 | 0.6087 | | | |
| 7 | Student assessment | Initial | 65 | 0.3846 | −4.914 | 155 | 0.000 |
| | | Secondary | 92 | 0.7500 | | | |
| 11 | Meeting with other subject/department faculty | Initial | 65 | 0.2154 | −3.895 | 155 | 0.000 |
| | | Secondary | 92 | 0.5109 | | | |
| 14 | Preparation of student progress reports | Initial | 65 | 0.1692 | −2.505 | 155 | 0.013 |
| | | Secondary | 92 | 0.3478 | | | |

**Table 4.** *Cont.*

| Item | After | Mentor | N | Mean | Statis. | gl | *p* |
|---|---|---|---|---|---|---|---|
| 4 | Production of learning materials and resources | Initial | 65 | 0.0769 | −2.697 | 155 | 0.008 |
|   |   | Secondary | 92 | 0.2391 |   |   |   |
| 6 | Design, development, or adaptation of instr, evaluation | Initial | 65 | 0.1077 | −2.107 | 155 | 0.037 |
|   |   | Secondary | 92 | 0.2391 |   |   |   |
| 7 | Student assessment | Initial | 65 | 0.2308 | −2.821 | 155 | 0.005 |
|   |   | Secondary | 92 | 0.4457 |   |   |   |
| 8 | Feedback to students | Initial | 65 | 0.1538 | −2.188 | 155 | 0.030 |
|   |   | Secondary | 92 | 0.3043 |   |   |   |

We now focus on those aspects of student teachers' performance that are observed and supervised by mentors during internships. As shown in Table 5, the majority of mentors observe and analyze all the key aspects of the student teacher's activity that may have an impact on their learning of the profession and their performance during the internship. This is a fact that contrasts with the results obtained previously with respect to the elements that work in the accompaniment process since it allows us to conclude that university and school mentors assume a traditional training model in which the student teacher would learn "by doing", where experience in a real context is the main source of learning. It is likely that mentoring conditions both in universities and in schools (lack of recognition and time) are limiting and condition the training model, but also behind this result are the existing difficulties in mobilizing active and collaborative processes of reflective learning.

**Table 5.** Descriptive statistics on the aspects to be developed by student teachers that are subject to observation and supervision by university and school mentors.

| Item | Observation Aspects | Mentor | N | % Cases |
|---|---|---|---|---|
| 1 | Compliance with standards or rules | University | 157 | 88.89 |
|   |   | School | 45 | 85.35 |
|   |   | Total | 202 | 86.10 |
| 2 | Lesson planning/teaching situations | University | 157 | 68.89 |
|   |   | School | 45 | 87.26 |
|   |   | Total | 202 | 83.20 |
| 3 | Selection or production of teaching materials | University | 157 | 64.44 |
|   |   | School | 45 | 80.25 |
|   |   | Total | 202 | 76.70 |
| 4 | Use of resources and technologies | University | 157 | 40.00 |
|   |   | School | 45 | 79.62 |
|   |   | Total | 202 | 70.80 |
| 5 | Design of teaching methodologies | University | 157 | 51.11 |
|   |   | School | 45 | 68.79 |
|   |   | Total | 202 | 64.90 |
| 6 | Assessment of student learning | University | 157 | 51.11 |
|   |   | School | 45 | 63.06 |
|   |   | Total | 202 | 60.40 |
| 7 | Making field records and notes | University | 157 | 55.56 |
|   |   | School | 45 | 45.86 |
|   |   | Total | 202 | 48.00 |
| 8 | Communication skills | University | 157 | 33.33 |
|   |   | School | 45 | 80.25 |
|   |   | Total | 202 | 69.80 |

**Table 5.** *Cont.*

| Item | Observation Aspects | Mentor | N | % Cases |
|---|---|---|---|---|
| 9 | Classroom interaction management skills | University | 157 | 48.89 |
| | | School | 45 | 83.44 |
| | | Total | 202 | 75.70 |
| 10 | Managing student motivation | University | 157 | 37.78 |
| | | School | 45 | 72.61 |
| | | Total | 202 | 64.90 |
| 11 | Managing group dynamics | University | 157 | 42.22 |
| | | School | 45 | 60.51 |
| | | Total | 202 | 56.40 |
| 12 | Key competencies (ICT, languages, entrepreneurship, teamwork, leadership, etc.) | University | 157 | 28.89 |
| | | School | 45 | 59.87 |
| | | Total | 202 | 53.00 |
| 13 | Reflection on practice | University | 157 | 86.67 |
| | | School | 45 | 75.80 |
| | | Total | 202 | 78.20 |
| 14 | Report writing | University | 157 | 75.56 |
| | | School | 45 | 34.39 |
| | | Total | 202 | 43.60 |
| 15 | Interaction with students | University | 157 | 64.44 |
| | | School | 45 | 81.53 |
| | | Total | 202 | 77.70 |
| 16 | Relationship with other professors and/or interns | University | 157 | 48.89 |
| | | School | 45 | 43.31 |
| | | Total | 202 | 44.60 |

Table 6 also reveals significant differences in aspects to be developed by student teachers that are subject to observation and supervision by mentors. Thus, there are nine aspects to which school mentors give more importance than university mentors in the observation and supervision of student teachers:

1. Lesson planning/teaching situations (sig = 0.004 < 0.05);
2. Use of resources and technologies (sig = 0.000 < 0.05);
3. Design of teaching methodologies (sig = 0.029 < 0.05);
4. Communication skills (sig = 0.000 < 0.05);
5. Classroom interaction management skills (sig = 0.000 < 0.05);
6. Management of student motivation (sig = 0.000 < 0.05);
7. Management of group dynamics (sig = 0.029 < 0.05);
8. Key competencies (sig = 0.000 < 0.05);
9. Interaction with students (sig = 0.015 < 0.05).

On the contrary, only in the writing of reports (sig = 0.000 < 0.05) do the university mentors give it more importance in the observation and supervision they conduct. We believe that these results could show the more than evident separation between university and professional cultures. The first is more focused on the formal aspects and the execution of the tasks and procedures designed for the learning of student teachers, and the second is truly focused on the professional teaching practice that, in our opinion, is the central object of the training process.

**Table 6.** Student's *t*-test on the aspects to be developed by student teachers that are subject to observation and supervision by university and school mentors.

| Item | | Mentor | N | Mean | Statis. | gl | *p* |
|---|---|---|---|---|---|---|---|
| 2 | Lesson planning/teaching situations | University | 45 | 0.6889 | −2.952 | 200 | 0.004 |
| | | School | 157 | 0.8726 | | | |
| 4 | Use of resources and technologies | University | 45 | 0.4000 | −5.501 | 200 | 0.000 |
| | | School | 157 | 0.7962 | | | |
| 5 | Design of teaching methodologies | University | 45 | 0.5111 | −2.205 | 200 | 0.029 |
| | | School | 157 | 0.6879 | | | |
| 8 | Communication skills | University | 45 | 0.3333 | −6.645 | 200 | 0.000 |
| | | School | 157 | 0.8025 | | | |
| 9 | Classroom interaction management skills | University | 45 | 0.4889 | −5.035 | 200 | 0.000 |
| | | School | 157 | 0.8344 | | | |
| 10 | Managing student motivation | University | 45 | 0.3778 | −4.506 | 200 | 0.000 |
| | | School | 157 | 0.7261 | | | |
| 11 | Managing group dynamics | University | 45 | 0.4222 | −2.196 | 200 | 0.029 |
| | | School | 157 | 0.6051 | | | |
| 12 | Key competencies (ICT, languages, entrepreneurship, teamwork, leadership, etc.) | University | 45 | 0.2889 | −3.781 | 200 | 0.000 |
| | | School | 157 | 0.5987 | | | |
| 14 | Report writing | University | 45 | 0.7556 | 5.205 | 200 | 0.000 |
| | | School | 157 | 0.3439 | | | |
| 15 | Interaction with students | University | 45 | 0.6444 | −2.452 | 200 | 0.015 |
| | | School | 157 | 0.8153 | | | |

The results also revealed the existence of significant differences in the aspects to be developed by the students in internships that are subject to observation and supervision by the school mentors depending on the educational levels to which they belong (initial or secondary). Thus, as shown in Table 7, mentors in Compulsory Secondary Education, Baccalaureate, and Vocational Training give more importance than those in Early Childhood and Primary to the following aspects:

1.  Lesson planning/teaching situations (sig = 0.022 < 0.05);
2.  Design of teaching methodologies (sig = 0.019 < 0.05);
3.  Assessment of student learning (sig = 0.019 < 0.05);
4.  Communication skills (sig = 0.012 < 0.05);
5.  Report writing (sig = 0.000 < 0.05).

**Table 7.** Student's *t*-test on the aspects to be developed by student teachers that are subject to observation and supervision by mentors depending on educational level.

| Item | | Mentor | N | Mean | Statis. | gl | *p* |
|---|---|---|---|---|---|---|---|
| 2 | Lesson planning/teaching situations | Initial | 65 | 0.8000 | −2.318 | 155 | 0.022 |
| | | Secondary | 92 | 0.9239 | | | |
| 5 | Design of teaching methodologies | Initial | 65 | 0.5846 | −2.375 | 155 | 0.019 |
| | | Secondary | 92 | 0.7609 | | | |
| 6 | Assessment of student learning | Initial | 65 | 0.5231 | −2.373 | 155 | 0.019 |
| | | Secondary | 92 | 0.7065 | | | |
| 8 | Communication skills | Initial | 65 | 0.7077 | −2.545 | 155 | 0.012 |
| | | Secondary | 92 | 0.8696 | | | |
| 14 | Report writing | Initial | 65 | 0.1538 | −4.447 | 155 | 0.000 |
| | | Secondary | 92 | 0.4783 | | | |

To conclude, the study also made it possible to identify some of the current limitations of the processes of supervision of the in-school placement, understood as learning processes that require greater horizontality between university and school mentors, permanent collaboration, and personalized monitoring of student teachers. The mentors pointed out the following aspects that, in their opinion, could improve the quality of the individualized follow-up of student teachers during the internships. They include:

1. More direct and fluid communication (22.3%);
2. A higher degree of collaboration (15.5%);
3. Protocols and tools to clarify the tasks and activities that students must carry out in the in-school placement (13.3%);
4. Instruments and resources that facilitate the individualized follow-up of interns throughout the process (9.5%);
5. Increasing the length of stay in the schools (7.7%);
6. Greater recognition of mentoring work (6.1%).

It should be noted that student mentors formulate the need for digital tools for networked learning that allow student teachers to systematically record their observations, promote reflective processes, give and receive feedback, communicate and collaborate, and share information.

## 4. Discussion

The study has reported on how the internship supervision process is currently developing and formulates some proposals to improve the individualized monitoring of student teachers from the point of view of their mentors (university and professional). The study has shown that the approaches taken by university and school mentors differ in multiple aspects. The differences found correspond to or explain the distance between academic and professional culture. Two different, scarcely connected worlds must necessarily collaborate in the initial training of teachers. We have seen the need for greater coordination between university and school mentors throughout the process. From the beginning, before the student teachers arrive at the placement schools, at which time the mentors should agree on the activities, the key elements on which the student teacher's learning will revolve, and the mechanisms to cooperate in the three phases prior to the placement and on closing.

We have also found that to ensure that the students' approach and observation of all the key aspects of the teaching function and life in schools, both mentors must agree and define them together, thus guaranteeing the progression of the training sequence and the integral nature of the experience.

The study has also shown that mentors, depending on their profile (academic or professional), acquire different importance in each of the phases, reducing the involvement of university mentors during their stay in the placement schools and of school mentors in the preparation and closing moments of the process. It is true that physical proximity to the student teacher implies a greater intensity in the training relationship. It is also true that for the triadic relationship to work, it is necessary to connect and link the three agents (university mentor, school mentor, and student teacher) in all stages of ITE.

In line with other studies (Gutiérrez-Provecho and López-Aguado 2012; Leránoz-Iglesias 2023; Ruiz-Gallardo et al. 2006), it can be said that the frequency of joint work between mentors and future teachers is conducted on a regular basis exclusively during their stay in the placement schools. This factor is determined by the contextualization and academic coordination of the subjects of the in-school placement and their link to official university degrees. In some cases, the school mentors do not have contact with the future teachers until the beginning of their stay at the placement schools themselves, which makes it impossible for them to carry out contacts and activities together beforehand. In terms of the type of activities, school mentors are more involved in working with student teachers in school practice on activities related to classroom dynamics (planning, teaching, materials), and university mentors are more involved in working with student teachers in

school practice on more organizational and formative aspects of the student teachers' own training plan.

Regarding the observation and supervision of the relevant aspects to be developed by student teachers, there are differences in performance between university and school mentors. University mentors focus their supervision on activities and processes related to reflection, reporting, and planning of student education, and, on the contrary, supervision of school mentors focuses on observation of those teaching activities and skills that student teachers carry out in their own classroom practice. This difference, in line with different studies (Bretones 2013; Gairín et al. 2019; Melgarejo 2014; Onrubia et al. 2020; Poveda et al. 2021), highlights the lack of collaboration and coordination between mentors in both contexts (university and professional) in the tasks to be performed by student teachers.

The absence of co-participation of both mentors in the follow-up of the student teacher can cause significant limitations and an imbalance in the mentoring process, generating isolated perceptions in the orientation and supervision of student teachers, which sometimes causes confusion and demotivation. According to some authors (González and Fuentes 2011; Rodríguez-Marcos et al. 2011; Tejada and Carvalho 2013), fluid dialogue and close cooperation, in addition to optimizing the quality of the accompaniment and competence development of student teachers, an enrichment for all parties, in the improvement of reflective practice, commitment, and teaching leadership. In line with some studies (Barriga 2016; Gallego-Arrufat and Cebrián-de-la-Serna 2018; González and Fuentes 2011; Hall et al. 2018; Leránoz-Iglesias 2023; Melgarejo 2014; Méndez 2012), it is necessary to rethink the in-school placement and make progress in key elements such as greater involvement of the educational administration, the definition of the essential characteristics that a placement mentor must possess and the incorporation of appropriate educational technologies to support collaboration, communication, and interaction in organizational processes, the management and development of the in-school placement, and all this, through a training plan where the agents involved are professionalized. These changes must be supported by a methodological model based on TPACK, shaped with new elements (Balladares-Burgos and Valverde-Berrocoso 2022), and adapted to the specific needs of the in-school placement. In this vein, some studies of mentoring experiences (Hixon and So 2009; Wells et al. 2023) show that they are improved when technology is used, encouraging the creation of shared experiences and reflections, as well as a fluid communication and relationship between both mentors.

The TPACK mechanism with the three types of knowledge, the curricular, the pedagogical and the technological in the intervention of the agents participating in the in-school placement leads to the design of a flexible, intelligent, versatile, and interoperable system that improves the dynamics of collaboration, regulates self-learning, conducts training assessment, and channels feedback to student teachers on an ongoing basis. In this sense, the piloting experiences of the EKT eLearning system (Egan et al. 2023; Fernández-Morante et al. 2022) demonstrate that through Educational Technologies and appropriate pedagogical methodologies it is possible to establish bridges between the school and university contexts, promote reflective learning processes and accompany student teachers through monitoring promoting the learning of the profession (Fernández-Morante and Bienzle 2023; Leránoz-Iglesias 2023; Leránoz-Iglesias et al. 2023).

**Author Contributions:** Conceptualization C.F.-M., B.C.-L. and M.M.L.-I.; methodology, C.F.-M., M.M.L.-I. and C.A.-P.; software, C.F.-M. and M.M.L.-I.; validation, M.M.L.-I., C.F.-M., C.A.-P. and B.C.-L.; formal analysis, M.M.L.-I., C.F.-M. and C.A.-P.; investigation, C.F.-M., B.C.-L., M.M.L.-I. and C.A.-P.; resources, B.C.-L. and C.F.-M.; data curation, C.A.-P. and M.M.L.-I.; writing—original draft preparation, M.M.L.-I., C.F.-M., B.C.-L. and C.A.-P.; writing—review and editing, M.M.L.-I., C.F.-M., B.C.-L. and C.A.-P.; visualization, C.A.-P. and M.M.L.-I.; supervision, C.F.-M., B.C.-L. and M.M.L.-I.; project administration C.F.-M. funding acquisition C.F.-M. All authors have read and agreed to the published version of the manuscript.

**Funding:** This research has received external funding from the Erasmus+ EKT—Educational Knowledge Transfer project (reference 612414-EPP-1-2019-1-ES-EPPKA2-KA).

**Institutional Review Board Statement:** The study was carried out in accordance with the deontological standards recognized by the Declaration of Helsinki (Hong Kong revision, September 1989) and in accordance with the recommendations of Good Clinical Practice of the EEC (document 111/3976/88 of July 1990) and the current Spanish legal regulations governing research.

**Informed Consent Statement:** Informed consent was obtained from all subjects involved in the study.

**Data Availability Statement:** Data are contained within the article.

**Conflicts of Interest:** The authors declare no conflict of interest.

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
