# Peer review of "Mentoring and Monitoring of Student Teachers in Their In-School Placements—The Case of the University of Santiago de Compostela"

_socsci, doi:10.3390/socsci13010017_

Round 1
Reviewer 1 Report
Comments and Suggestions for Authors
Dear authors,
I found the manuscript compelling and well written. I have some comments that you shoud address:
1) The use of acronyms is confusing. Please, define the acronym at the very first appearance and then later on, use the acronym.
2) In the literature, the student teacher use is not correct. Should be pre-service teacher.
3) In the abstract, the section 4) is confusing, please rewrite it, introducing your proposal.
4) In all the tables you use commas. You should use points.
5) Figure 5 needs information of the axes. In addition, it is confusing seeing 14 elements when they are not previously described.
6) I am missing more information on the scheme you are using. How the elements of Table 1 are associated to your educational approach. Also, how the mentors are prepared or are aware of the elements. Who is considering to promote the 14 elements within the pre-service teachers. I guess you need to introduce a new paragraph introducing the instructional approach for mentoring.
7) The conclusion, in it present form, still looks as a piece of discussion. Please do not use references in the conclusions and try to summarize the results in a more concise way.
Author Response
Response to reviewer 1 Comments
I found the manuscript compelling and well written. I have some comments that you shoud address:
1) The use of acronyms is confusing. Please, define the acronym at the very first appearance and then later on, use the acronym.
The use of ACRONYMS has been revised and corrected according to the evaluator's criteria. Amendments highlighted in yellow
2) In the literature, the student teacher use is not correct. Should be pre-service teacher.
We accept and appreciate the reviewer's suggestion, but we understand that there are many references in the literature to the concept of "student teachers" and that they are equally valid as the suggested concept or others that would also be appropriate, such as beginning teacher. Leading authors in the field such as Zeichnner & Liston, Feiman-Nemser, Popkewitz, Wayne & Woolfolk, among others, have traditionally used it in their publications and research. The EKT competitive research project in which this publication is framed focuses on Initial Teacher Education and for this reason we decided to use this concept in its design and implementation, obtaining a favourable evaluation in its selection and final evaluation. We are grateful for the consideration, which is undoubtedly interesting, but we understand that the concept is valid and for coherence with the research in which it is framed and the foreseen publications we prefer to keep it.
3) In the abstract, the section 4) is confusing, please rewrite it, introducing your proposal.
The paragraph has been rewritten to make it clearer. Changes highlighted in yellow
4) In all the tables you use commas. You should use points.
Following the reviewer's instructions, commas have been changed in all the tables of the manuscript and replaced by full stops.
5) Figure 5 needs information of the axes. In addition, it is confusing seeing 14 elements when they are not previously described.
The figure refers to the dimensions presented on the two axes (percent and Key elements), however, due to the high amount of information that the variables analysed represent, it has been decided not to include the legends referring to each variable, but to include their corresponding number. These legends can be easily identified in the following table nº1. The graph is intended to visually show the different values at each stage (before, during and after) in order to quickly detect those not addressed in any of them. For this reason, it is understood that the figure should not incorporate more information than necessary, as it is complemented by the following table (table 1).
6) I am missing more information on the scheme you are using. How the elements of Table 1 are associated to your educational approach. Also, how the mentors are prepared or are aware of the elements. Who is considering to promote the 14 elements within the pre-service teachers. I guess you need to introduce a new paragraph introducing the instructional approach for mentoring.
Precisely, the research shows that there is no shared model or coordinated strategy between faculties and schools to address the practicum. As indicated and clarified in yellow at the beginning of the results report, table 1 results from the consensus among the 5 European ITE institutions participating in the project on the key and necessary elements for reflective learning and aims to assess the extent to which they are addressed. As a consequence of this analysis, the project subsequently designs a methodology and an e-learning system aligned with the needs identified. The training model is built on the basis of the diagnosis and subsequently tested. This publication refers to the needs analysis phase.
7) The conclusion, in it present form, still looks as a piece of discussion. Please do not use references in the conclusions and try to summarize the results in a more concise way.
Indeed, the conclusion section is brief. The intention was to emphasise the importance of the application of technology by creating an e-learning system for ISP. Since SS journal states that the conclusion section is optional and "may be added if the discussion is unusually long or complex", but this is not the case, taking into account the reviewers' comments, it was decided to delete the section.
Thank you for the valuable contributions which will certainly help to improve the manuscript.
Reviewer 2 Report
Comments and Suggestions for Authors
Well done on the work to date on this interesting paper.
Here are some suggestions:
Evaluate the use of the word "trainee" - teacher education v teacher training are very different concepts, so it is important to position yourself relation to this and watch the language used accordingly.
Sentrence from line 26 to 31 is very long and may benefit by being separated into two sentences.
Rephrase line 23 to 26. Suggest something like
The research is an integral component of the accredited university programs that prepare individuals for the teaching profession at the University of Santiago de Compostela. Its objective is to conduct an examination of the procedures involved in individually monitoring student teachers during their internships, specifically from the perspective of their mentors.
The paragraph from line 33 to 91 is very long and the paper would benefit by this being seprated into at least two paragraphs where appropriate.
References
The paper would benefit with more up to date and internaitonal references regarding mentoring in teacher education.
Informatin such as that in line 311 to 317 may be clearer in a different format, consider a list or a table?
In text referencing - e.g. line 504 - full list of authors required for the reference - wath this throughout.
Some points made early inthe discussion piece would benefit from links to the literature as is done in the later section.
The conclusion is very brief and emphasises the TPACK model that onlly appears briefly in the paper desipite also being mentioned in the abstract?
The use rationale for TPACK should be more clearly exlicated if it so important to the paper - the only real mention of it is in line 92 and 452 - so a suggestion is a short "conceptual framework" section to explain the importance of the TPACK model to the reseach.
Besr of luck with the revisions.
Comments on the Quality of English Language
The quality of English language is good. Some proofing re length of sentences and paragraphs would enhance the paper.
Author Response
Response to reviewer 2 Comments
Here are some suggestions:
Evaluate the use of the word "trainee" - teacher education v teacher training are very different concepts, so it is important to position yourself relation to this and watch the language used accordingly.
The evaluator's precision is appreciated and the use of the concept is revised and modified. "Student teacher trainee" is replaced by "student teacher in school practice". The changes highlighted in green include a revision of the title of the article following the reviewer's indications.
Sentrence from line 26 to 31 is very long and may benefit by being separated into two sentences.
The wording has been corrected to make it clearer. Changes highlighted in green are indicated
Rephrase line 23 to 26. Suggest something like
The research is an integral component of the accredited university programs that prepare individuals for the teaching profession at the University of Santiago de Compostela. Its objective is to conduct an examination of the procedures involved in individually monitoring student teachers during their internships, specifically from the perspective of their mentors.
The drafting suggestion is gratefully acknowledged and the paragraph is amended as indicated, which undoubtedly improves and clarifies the manuscript as a result of the research carried out.
The paragraph from line 33 to 91 is very long and the paper would benefit by this being seprated into at least two paragraphs where appropriate.
There are indeed a lot of ideas in one paragraph and they are reorganised into two.
References
The paper would benefit with more up to date and internaitonal references regarding mentoring in teacher education.
Updated and international references on mentoring in teacher education are added as directed by the evaluator.
Information such as that in line 311 to 317 may be clearer in a different format, consider a list or a table?
we are aware of the complexity of the dimension analysed and the variables involved. In any case, we consider that, as referred to in the introductory paragraph to the aforementioned lines, the complete list is visible in table 4 and the complete table with the data discussed in lines 311-317. In the drafting of the manuscript, we considered making coloured masks in the tables to draw attention to the results highlighted, but we did not do so because the table formatting rules of the journal do not allow it.
In text referencing - e.g. line 504 - full list of authors required for the reference - wath this throughout.
This is indeed an error. The 4 references in which the authors were incomplete have been corrected.
Some points made early inthe discussion piece would benefit from links to the literature as is done in the later section.
The conclusion is very brief and emphasises the TPACK model that onlly appears briefly in the paper desipite also being mentioned in the abstract?
Indeed, the conclusion section is brief. The intention was to emphasise the importance of the application of technology by creating an e-learning system for ISP. Since SS journal states that the conclusion section is optional and "may be added if the discussion is unusually long or complex", but this is not the case, taking into account the reviewers' comments, it was decided to delete the section.
The use rationale for TPACK should be more clearly exlicated if it so important to the paper - the only real mention of it is in line 92 and 452 - so a suggestion is a short "conceptual framework" section to explain the importance of the TPACK model to the reseach.
Following the reviewer's indications, the explanation of TPACK's contribution to the conceptual framework is completed.
Best of luck with the revisions.
Thank you for the valuable contributions which will certainly help to improve the manuscript.
Round 2
Reviewer 1 Report
Comments and Suggestions for Authors
Dear authors,
Figure 1 needs to have information in the vertical axes. This is mandatory for any graph.
I still would prefer pre-service teachers, since is the most common term.
Author Response
A further reference to the axis is included following the indications of the evaluator. As indicated in the previous answer, for consistency with the concepts used in all the publications derived from the EKT project and in the application itself approved by the European Union, the term "student teacher" we prefer to maintained.